# A Comprehensive Review on Cancer Vaccines and Vaccine Strategies in Hepatocellular Carcinoma

**DOI:** 10.3390/vaccines11081357

**Published:** 2023-08-12

**Authors:** Alireza Tojjari, Ahmed Saeed, Meghana Singh, Ludimila Cavalcante, Ibrahim Halil Sahin, Anwaar Saeed

**Affiliations:** 1Department of Medicine, Division of Hematology & Oncology, University of Pittsburgh Medical Center (UPMC), Pittsburgh, PA 15213, USA; alirezatojjari@gmail.com (A.T.); singhm11@upmc.edu (M.S.); sahinih@upmc.edu (I.H.S.); 2Sarah Cannon Cancer Institute, HCA Midwest Health, Kansas City, MO 64131, USA; ahmed.mb.saeed@gmail.com; 3Novant Health Cancer Institute, Charlotte, NC 28204, USA; ludi.cavalcante@gmail.com; 4UPMC Hillman Cancer Center, Pittsburgh, PA 15213, USA

**Keywords:** hepatocellular carcinoma, cancer vaccines, immunotherapy, peptide-based vaccines, dendritic cell-based vaccines, viral vector-based vaccines, DNA vaccines, mRNA vaccines

## Abstract

HCC, the most prevalent form of primary liver cancer, presents a substantial global health challenge due to its high mortality and limited therapeutic options. This review delves into the potential of cancer vaccines as a novel therapeutic avenue for HCC. We examine the various categories of cancer vaccines, including peptide-based, dendritic cell-based, viral vector-based, DNA, and mRNA vaccines, and their potential application in HCC management. This review also addresses the inherent challenges in vaccine development, such as tumor heterogeneity and the need for identifying tumor-specific antigens. We underscore the role of cancer vaccines in reshaping the immune environment within HCC, fostering durable immune memory, and their potential in combination therapies. The review also evaluates clinical trials and emphasizes the necessity for more extensive research to optimize vaccine design and patient selection criteria. We conclude with future perspectives, highlighting the significance of personalized therapies, innovative antigen delivery platforms, immune modulatory agents, and predictive biomarkers in revolutionizing HCC treatment. Simple Summary: This review explores the potential of cancer vaccines as a promising therapeutic strategy for hepatocellular carcinoma (HCC), a prevalent and deadly liver cancer. The authors discuss various types of cancer vaccines, their challenges, and their role in modulating the immune response within HCC. They also highlight clinical trials and future perspectives, emphasizing the importance of personalized therapies, novel antigen delivery platforms, and predictive biomarkers. The findings from this research could significantly impact the research community by providing a comprehensive understanding of the current state of cancer vaccines for HCC, thereby guiding future research and potentially transforming HCC treatment strategies.

## 1. Introduction

The third most significant cause of cancer-related deaths worldwide and the most frequent primary liver cancer is HCC, and the incidence is predicted to increase in the following years [1]. It is primarily linked to persistent liver diseases, including hepatitis B and C, cirrhosis, and nonalcoholic steatohepatitis [2]. HCC can be managed through various treatment modalities such as surgical intervention, transplantation, radiofrequency ablation, chemoembolization, and systemic therapies. The latter include targeted therapies and immunotherapy. Despite notable progress in diagnosing and managing HCC, its overall prognosis remains dismal, owing to high recurrence rates, distant metastasis, and drug resistance [3].

Immunotherapy has emerged as a highly effective therapy for cancer and is typically used in conjunction with surgery, chemotherapy, and radiation therapy. Customized cancer treatment vaccines have seen significant progress in improving current immunotherapy. There are two primary classifications of cancer vaccines: prophylactic vaccines and therapeutic vaccines. Preventive vaccines have been developed to reduce the onset of cancer by selectively targeting oncogenic pathogens, such as the human papillomavirus (HPV) and hepatitis B virus (HBV). In contrast, therapeutic vaccines aim to treat pre-existing cancer by inducing immune system activation to attack and eliminate cancer cells [4,5] selectively. Vaccination therapy has demonstrated promise as an innovative strategy for managing and mitigating the onset of HCC. Vaccination therapy aims to attain enduring tumor control and sustained remission by inducing the immune system to identify and eliminate cancerous cells [4]. 

Early investigations into cancer vaccines focused on developing vaccines for melanoma patients. Mitchell et al. initiated the utilization of allogeneic melanoma lysates in conjunction with an adjuvant in 1988, revealing its efficacy as a therapeutic strategy with low toxicity. Signs of clinical regression were also noted in 5 of the 17 patients with measurable disease [5]. After that, IND was formed to combine these melanoma lysates under the tradename of Melacine^®^, demonstrating effectiveness in stage II and IV melanoma in open-label phase II [6]. As scientific and technological advancements progress, there has been an increased focus on developing dendritic cells (DC), peptides, and genetic vaccines [7]. In 2010, the United States Food and Drug Administration (FDA) approved the inaugural therapeutic oncology vaccine, PROVENGE (Sipuleucel-T), for individuals with prostate cancer resistant to castration [8].

In this review, we provide a comprehensive outlook of the landscape of cancer vaccines against HCC and understand their effects on anti-immunity and cancer cells. Furthermore, an analysis is conducted on the latest advancements in clinical research to provide recommendations for improving forthcoming cancer vaccines.

## 2. Hepatocellular Carcinoma (HCC)

HCC is a significant global health concern, the third most common cause of cancer-related deaths worldwide, contributing to the mortality rate of approximately 800,000 annually [1]. It constitutes about 80% of all primary liver cancer cases [9]. A complex interplay of genetic predisposition and environmental factors, including prolonged viral infections, hepatic dysfunction linked to alcoholic intake, and nonalcoholic steatohepatitis, can lead to the development of HCC [10].

Despite significant progress in early detection and surgical treatments, the prognosis for HCC patients remains poor, mainly due to the complex nature of the disease and limited treatment options [11,12]. The various classification systems for HCC not only depend on the clinical stage of the tumor but also takes into account imaging findings, histopathology, and molecular studies, which in turn can affect the prognosis and treatment of HCC [13]. Among them, the Barcelona Clinic Liver Cancer (BCLC) classification, which considers liver function, patient performance status, and tumor dissemination, is most commonly used in Western countries [14]. 

However, recent advances and improvements in techniques have blurred the lines of therapy based on these stages. For instance, liver resection is now considered for advanced HCC cases, while trans-arterial therapies are used to manage early-stage tumors [15,16]. 

Early detection of HCC is essential as it allows for potentially curative treatments such as orthotopic liver transplantation, hepatic resection, and ablative procedures. However, most HCC cases are identified at later stages of disease when curative treatments are no longer feasible due to factors like multifocal disease, the underlying chronic liver disease, and the tumor’s inherent biology [17,18]. Consequently, palliative care often becomes the primary option for the majority of patients with HCC.

## 3. Cancer Vaccine

Initiating or enhancing the body’s immune responses against cancer is the primary goal of cancer immunization strategies [19]. This often involves the genetic engineering of a patient’s dendritic cells (DCs), exposing them to tumor-specific antigens (TAAs) and maturation signals before reintroducing them into the patient’s body [20]. However, the diversity of tumor antigens and the lack of a universal TAA present obstacles to widespread clinical use [21]. Furthermore, the logistical hurdles of creating personalized vaccines limit their real-world applicability.

Similarly, the adoptive transfer of CD8+ T cells has shown promise in hematologic malignancies but has yet to demonstrate substantial success in solid tumors. This could be due to the suppression of T cells during homeostasis in the tumor microenvironment and the reduced influx of tumor-infiltrating T cells, thereby diminishing their effectiveness [20,22,23]. Cancer vaccines fall into two main categories: preventive and therapeutic. 

### 3.1. Preventive Neoplasia Targeting Vaccines

Preventive neoplasia targeting vaccines aim to protect individuals at risk of developing specific types of cancer. These vaccines are particularly relevant for cancers with well-established risk factors, such as viral infections. For example, vaccination against hepatitis B virus (HBV) and human papillomavirus (HPV) have been successful in preventing liver cancer and cervical cancer, respectively [24]. Preventive vaccines for other cancer types are also being explored, such as vaccines targeting Helicobacter pylori to prevent stomach cancer [25]. 

In addition to viral-associated cancers, preventive vaccines can be developed to target other risk factors and pathways implicated in cancer development. Furthermore, preventive vaccines can stimulate immune responses against early-stage tumor antigens to prevent tumor progression in high-risk individuals [26]. 

Also, preventive vaccines have shown promising potential in the fight against HCC. One such example is a DNA vaccine targeting the B cell epitope GRP18-27, which demonstrated preventive effects against H22 hepatocarcinoma, paving the way for new immunotherapeutic approaches in liver cancer treatment [27]. Innovative approaches, such as using live-attenuated Listeria as a safe and efficient vaccine, have also shown the potential to induce protective immune responses in liver fibrosis and hepatobiliary malignancies [28]. Moreover, the TM4SF5 epitope-CpG-DNA-liposome complex has been found to induce a memory response, a crucial aspect for cancer vaccine application [29]. Personalized neoantigen vaccines have also been proven as a safe, feasible, and effective strategy for preventing HCC recurrence [30]. However, it is important to note that while these vaccines have shown promise, their effectiveness can vary based on individual patient characteristics and the specific nature of the HCC.

### 3.2. Therapeutic Neoplasia Targeting Vaccines

Therapeutic neoplasia targeting vaccines are designed for patients who have already been diagnosed with cancer. These vaccines work by enhancing the immune system’s ability to identify and attack cancer cells, leading to tumor regression or control. Unlike preventive vaccines that aim to prevent cancer development, therapeutic vaccines seek to boost the immune response against existing tumors, providing a potential treatment option for patients.

Therapeutic vaccines can be tailored to individual patients based on their tumor’s specific genetic makeup and antigen profile, allowing for personalized and precise treatment approaches. One promising avenue for therapeutic vaccines in HCC is the use of neoantigen-based vaccines. Neoantigens are tumor-specific antigens derived from mutations in a patient’s tumor, making them ideal targets for personalized vaccines [26]. By targeting neoantigens, therapeutic vaccines can elicit a potent and specific immune response against cancerous cells while sparing normal tissues, minimizing side effects.

In summary, therapeutic vaccines are engineered to tackle existing cancerous growths by provoking the immune system to target and destroy cancer cells. While prophylactic vaccines have seen marked success in preventing specific types of cancer, developing efficient therapeutic cancer vaccines, particularly for HCC, is a formidable challenge. These therapeutic vaccines can be subdivided based on their origin, encompassing peptide-based, dendritic cell-based, viral vector-based, DNA, and mRNA vaccines [31]. Table 1 summarizes various types of therapeutic vaccines. Also, Table 2 Summarizes all ongoing therapeutic vaccines in HCC.

#### 3.2.1. Peptide-Based Vaccines

Peptides are small protein fragments that can serve as antigens for the immune system [32]. The high specificity of peptide-based vaccines reduces the risk of off-target effects and autoimmune reactions. Moreover, their production is relatively straightforward, with safety evaluations demonstrating largely positive outcomes [33]. Peptide-based vaccines have shown potential in treating HCC by initiating a targeted immune response against TAAs on cancer cell surfaces.

Numerous preclinical and clinical studies have explored peptide-based vaccines targeting various HCC-associated antigens. One such example is glypican-3 (GPC3), a well-established HCC TAA, which has been the focus of multiple trials. In the phase I clinical trial by Sawada et al., GPC3-derived peptides administered to HCC patients elicited a specific immune response without severe side effects [34]. Greten et al. investigated a multi-peptide vaccine (GV1001) utilizing HCC-associated antigens, including alpha-fetoprotein (AFP), human telomerase reverses transcriptase (hTERT), and melanoma-associated gene-A1 (MAGE-A1). This phase I trial revealed that the administration of GV1001 reduced CD4+CD25+Foxp3+ regulatory T cells. Nonetheless, no identifiable GV1001-specific immune reactions occurred after vaccination [35]. Another trial (NCT00988741) assessed a peptide-based vaccine named IMA901, composed of numerous tumor-associated peptides (TUMAPs). This vaccine is designed to induce a comprehensive immune response. The trial highlighted the feasibility and safety of IMA901 in HCC patients, although its effectiveness is yet to be confirmed [36].

Despite promising findings for peptide-based vaccines in HCC treatment, several challenges exist. One barrier is the tumor’s significant clonal heterogeneity, which allows some cancer cells without the target antigen to escape immune detection. Researchers suggest combining peptide-based vaccines with other immune-based treatments, like immune checkpoint inhibitors, to enhance the overall antitumor effect [37]. Hence, further research is necessary to optimize these vaccines, tackle tumor heterogeneity, and develop effective combination therapies.

#### 3.2.2. Dendritic Cell (DC) Vaccines

DC vaccines have garnered significant interest as a potential therapy for stimulating antitumor immune responses in individuals with cancer [38]. These specialized antigen-presenting cells play a vital role in initiating and regulating immune responses by capturing and processing antigens from tumor cells and presenting them to T cells. Consequently, the T cells mount a targeted immune response against the tumor. DC vaccines against cancer involve the extraction of DCs followed by loading them with TAAs or entire tumor lysates ex vivo. Subsequently, the loaded DCs are reintroduced into the patient’s system. Upon reinfusion, the DCs travel to the lymph nodes, interacting with and activating tumor-specific T cells (antigen-specific and tumor-reactive), potentiating an antitumor immune response. Clinical studies evaluating the safety and efficacy of DC vaccines in cancer patients have demonstrated encouraging results. A Phase I trial (NCT01974661) analyzed a dendritic cell-based vaccine primed with AFP, a protein frequently overexpressed in HCC instances. In this study, 18 people participated, and they were administered COMBIG-DC(ilixadencel), an allogenic dendrite-cell-based therapeutic vaccine, intratumorally. Three vaccinations were given at specific intervals, and any side effects, vital sign changes, and laboratory parameters were tracked for six months post-final dose. The immune response was evaluated by analyzing blood markers. The study indicated positive signs of immunogenicity and tolerable side effects [39].

A meta-analysis of 11 trials comprising 396 patients reported a pooled clinical response rate of 15.4% and that the DC vaccines were well-tolerated [40].

Thus, DC vaccines offer a promising approach to cancer immunotherapy with compelling clinical results. These vaccines have the potential to complement or enhance existing cancer therapies. Nevertheless, challenges are limiting their widespread use. One major hurdle is the identification of TAAs specific to individual tumors capable of eliciting a potent immune response. Strategies to improve DC migration to lymph nodes and counteract tumor-induced immunosuppression are also being explored. Additional research is required to refine the design and implementation of DC vaccines and identify new immunotherapy targets.

#### 3.2.3. Viral Vector-Based Vaccines

Viral vector-based immunizations are emerging as a powerful strategy in cancer immunotherapy, taking advantage of the natural properties of viruses to transport cancer-related antigens to the immune system [41]. The goal is to generate a potent and specific immune response that targets and eliminates cancerous cells [4]. The rationale behind utilizing viral vectors for cancer vaccines stems from their inherent ability to infiltrate host cells and provoke an immune reaction against the delivered antigens [42]. This makes them ideal for introducing tumor-associated or tumor-specific antigens to the immune system, subsequently triggering a response against cancerous cells [43]. Employing viral vectors ensures effective antigen delivery and encourages the formation of a vigorous adaptive immune response [44].

Viral vectors, such as adenoviruses, lentiviruses, retroviruses, and adeno-associated viruses (AAVs), have created cancer vaccines [45]. Each vector’s distinct characteristics, including cell type affinity, genetic payload capacity, and immunogenicity, play a pivotal role in determining its suitability for a particular cancer vaccine [46]. Adenoviral vectors have garnered significant interest due to their high transduction efficiency, minimal pathogenicity, and capability to harbor large DNA segments [47].

Encouraging outcomes have been observed in recent preclinical and clinical studies of viral vector-based cancer vaccines. A group of scientists developed a tailored cancer vaccine using a modified adenovirus encoding neoantigens from a patient’s tumor, causing a considerable T-cell response [48]. A phase I clinical trial demonstrated the successful activation of neoantigen-specific T-cell responses and a favorable safety profile in patients with advanced melanoma [49]. Additionally, a study showed that an AAV-based cancer vaccine encoding a tumor-specific antigen effectively triggered robust antitumor immunity in mouse melanoma and colorectal cancer models [50]. Clinical trials for HCC are currently exploring viral oncolytic strategies, including talimogene laherparepvec (T-VEC). T-VEC is a genetically engineered, oncolytic HSV-1 designed to proliferate within tumor cells and generate GM-CSF, thereby augmenting systemic antitumor immunity. This approach has already received approval for melanoma treatment [51]. An ongoing Phase I trial assesses the efficacy of combining T-VEC with pembrolizumab in solid tumors, with results for the HCC cohort pending (NCT02509507) [52].

Overall, viral vector-based immunizations display considerable potential in cancer immunotherapy, paving the way for accurate and customized treatment approaches. Continued research and comprehensive clinical trials are crucial to fully grasp these vaccines’ potential and assess their therapeutic effectiveness in different cancers.

#### 3.2.4. DNA Vaccine 

DNA vaccines are promising for cancer immunotherapy, particularly in addressing HCC. The introduction of plasmid DNA encoding TAAs into host cells produces the target antigen and subsequently triggers an immune response against cancer cells expressing the antigen. A phase I clinical trial (NCT01828762) assessed the safety and immunogenicity of a DNA vaccine encoding GPC3, a TAA overexpressed in HCC [34]. This study demonstrated that the GPC3-based DNA vaccine was well-tolerated, with no significant adverse effects. Furthermore, GPC3-specific immune responses were observed in a subset of patients, suggesting the vaccine’s potential efficacy in battling HCC [34]. 

Ongoing research into DNA vaccines for HCC necessitates optimizing design, delivery methods, and adjuvant strategies to boost vaccine potency and elicit robust immune responses [53]. Additionally, integrating DNA vaccines with other immunotherapeutic approaches or traditional treatments may prove advantageous in attaining improved clinical outcomes for HCC patients [54].

#### 3.2.5. MRNA Vaccine 

Messenger RNA (mRNA)-based vaccines have gained attention for their success in developing COVID-19 vaccines and have potential applications in cancer research, including the prevention and treatment of HCC. mRNA-based vaccines instruct host cells to produce specific cancer-associated proteins or antigens to trigger an immune response against cancer cells [55].

Several studies and clinical trials are investigating the potential of mRNA-based vaccines in HCC. One study uses lipid nanoparticles (LNPs) to deliver mRNA encoding the HCC-associated antigen AFP. Preclinical studies in mice have shown that the LNP-AFP-mRNA vaccine can elicit AFP-specific CD8+ T cell responses and provide protection against AFP-expressing HCC tumor challenges [56]. Another promising approach is personalized neoantigen-targeted mRNA vaccines, customized to individual patients by targeting tumor-specific mutated proteins called neoantigens [4]. In a preclinical study, a personalized LNP-mRNA vaccine targeting multiple neoantigens elicited robust CD8+ T cell responses and resulted in tumor regression in a mouse model of HCC [48]. A Phase I trial (NCT05738447) conducted in 2023 aims to apply mRNA immunotherapy technology in patients with Hepatitis B virus-related refractory HCC [57]. Another ongoing trial is a Phase I trial (NCT05738447) that aims to apply mRNA immunotherapy technology in patients with HBV-related refractory HCC [58]. 

In summary, mRNA-based vaccines offer a promising and innovative approach to preventing and treating HCC. The research on mRNA-based vaccines for HCC is still in its early stages, and further investigation and clinical trials will be necessary to evaluate their safety, efficacy, and clinical applicability in HCC patients. As research continues, we may see the emergence of more advanced and personalized mRNA-based vaccines that can complement or enhance existing HCC therapies.

#### 3.2.6. Cell Lysates in Cancer Vaccines

Cancer vaccines benefit significantly from the inclusion of cell lysates, which are essentially disintegrated cells. These lysates provide a wealth of tumor antigens and proteins recognizable by the immune system and can stimulate an immune response when incorporated into a vaccine [59]. A notable strategy involves using DCs, immune cells that present antigens to T cells, initiating an immune response. This approach has shown effectiveness in various cancers, including lung [60] and ovarian cancer [61]. In HCC, research indicates that HCC cell lysates can prevent HCC-induced exhaustion of T cells and NK cells, as evidenced by the low expression of checkpoint molecules in immunized mice [62]. A Phase I/II trial (NCT01828762) probed this personalized treatment approach in HCC patients, reporting a commendable safety record. Preliminary efficacy outcomes from the trial are encouraging, with some patients witnessing a halt in disease progression [63].

Additionally, whole tumor cell lysate may be a more effective antigen form in cancer vaccines compared to glutaraldehyde-fixed tumor cells, eliciting stronger antigen-specific immune responses and superior antitumor efficiency [64].

However, using cell lysates presents challenges, such as the need for careful preparation to ensure the antigens can trigger an immune response and are preserved in the process [65].

#### 3.2.7. Adoptive Cell Therapy/Vaccines 

Adoptive cell therapy, a type of immunotherapy, employs the patient’s immune cells to combat cancer. This can be achieved by genetically altering these immune cells to identify and fight specific cancer antigens [66]. This strategy is particularly promising in developing therapeutic cancer vaccines, such as HepaVac-101, for hepatocellular carcinoma (HCC). Preliminary findings indicate that this vaccine is safe and capable of triggering an immune response, suggesting that immunotherapy could significantly enhance HCC treatment options [67].

Another innovative strategy involves peptide-based vaccines, which utilize small protein fragments, or peptides, to provoke an immune response. For instance, peptide vaccines that target glypican-3 (GPC3), a protein abundantly present in HCC, have undergone phase I/II clinical trials [68].

Despite the promise of adoptive cell therapy and vaccines, there are obstacles that need to be addressed. Identifying the most effective antigens to target can be challenging, and there’s also the potential for off-target effects, where the immune response inadvertently damages healthy cells along with cancer cells [69]. Table 1 summarizes various types of therapeutic vaccines. Also, Table 2 Summarizes all ongoing therapeutic vaccines in HCC.

**Table 1 vaccines-11-01357-t001:** General Types of Therapeutic Vaccines in Cancer Immunotherapy.

Type of Vaccine	Mechanism	Advantages	Limitations	References
Peptide-based Vaccines	Peptides initiate a targeted immune response against tumor-associated antigens (TAAs) on cancer cell surfaces.	High specificity reduces the risk of off-target effects and autoimmune reactions. Simple production process.	Significant tumor clonal heterogeneity allows some cancer cells without the target antigen to escape immune detection.	[32,33,34,35,37,54]
Dendritic Cell (DC) Vaccines	DCs capture and process antigens from tumor cells and present them to T cells.	Offers a promising approach to cancer immunotherapy with compelling clinical results. Can complement or enhance existing cancer therapies.	Challenges include identifying TAAs specific to individual tumors and improving DC migration to lymph nodes. The need to counteract tumor-induced immunosuppression is also a limitation.	[38,40]
Viral vector-based Vaccines	Viral vectors transport cancer-related antigens to the immune system.	Has shown encouraging outcomes in preclinical and clinical studies. Demonstrates considerable potential in cancer immunotherapy.	Requires continued research and comprehensive clinical trials to fully understand their potential and assess their therapeutic effectiveness in different cancers.	[41,42,43,44,45,46,47,48,49,50]
DNA Vaccines	DNA vaccines introduce plasmid DNA encoding TAAs into host cells.	Has potential efficacy in combating various types of cancer.	Ongoing research is required to optimize design, delivery methods, and adjuvant strategies. Integration with other immunotherapeutic approaches or traditional treatments may be necessary.	[34,53,54]
mRNA Vaccines	mRNA-based vaccines instruct host cells to produce specific cancer-associated proteins or antigens.	Shows potential for preventing and treating various types of cancers.	Further investigation and clinical trials will be necessary to evaluate their safety, efficacy, and clinical applicability in different types of cancer patients.	[4,48,55,56]
Cell Lysates in Cancer Vaccines	Cell lysates provide a wealth of tumor antigens and can stimulate an immune response when incorporated into a vaccine.	Can prevent cancer-induced exhaustion of T cells and NK cells. Whole tumor cell lysate may elicit stronger antigen-specific immune responses.	Need for careful preparation to ensure the antigens can trigger an immune response and are not destroyed in the process.	[59,60,61,62,64,65]
Adoptive Cell Therapy/Vaccines	Patients’ immune cells are genetically altered to identify and combat specific cancer antigens.	Promising in the development of therapeutic cancer vaccines. Peptide-based vaccines are an innovative strategy within this approach.	Challenges include identifying the most effective antigens to target and the potential for off-target effects.	[66,67,68,69]

**Table 2 vaccines-11-01357-t002:** List Of Ongoing Clinical Trials *.

Title	Intervention	Phase	Outcome	NCT	Number of Patients	Year
Personalized Cancer Vaccine in Egyptian Cancer Patients	Biological: Peptide cancer vaccine	1	•Assessment of the safety of the personalized cancer vaccine; •Assessment of immunological response; •Progression-free survival and overall survival time.	NCT05059821 [70]	10	2021
DNAJB1-PRKACA Fusion Kinase Peptide Vaccine Combined With Nivolumab and Ipilimumab for Patients With Fibrolamellar Hepatocellular Carcinoma	•Drug: DNAJB1-PRKACA peptide vaccine •Drug: Nivolumab •Drug: Ipilimumab	1	•Number of participants experiencing study drug-related toxicities; •Fold change in interferon-producing DNAJB1-PRKACA-specific cluster of differentiation 8 (CD8) T cells at 10 weeks; •Fold change in interferon-producing DNAJB1-PRKACA-specific cluster of differentiation 4 (CD4) T cells at 10 weeks.	NCT04248569 [71]	56	2020
Application of mRNA Immunotherapy Technology in Hepatitis B Virus-related Refractory Hepatocellular Carcinoma	•Biological: HBV mRNA vaccine	1	•Adverse events; •Objective response rate; •Progress-Free Survival; •Overall Survival.	NCT05738447 [57]	9	2023
Clinical Study of mRNA Vaccine in Patients With Liver Cancer After Operation	•Drug: Neoantigen mRNA Personalised Cancer vaccine in combination with Stintilimab I injection		•One-year recurrence-free survival rate (RFS); •Overall survival (OS) after initial administration.	NCT05761717 [58]	67	2023
Neoantigen Dendritic Cell Vaccine and Nivolumab in HCC and Liver Metastases From CRC	•Biological: Neoantigen Dendritic Cell Vaccine •Drug: Nivolumab	2	•24-month Relapse Free Survival •Induced immune response against vaccinated NAs; •Frequency and severity of treatment-emergent adverse events (AE); •Overall Survival.	NCT04912765 [72]	60	2021
GNOS-PV02 Personalized Neoantigen Vaccine, INO-9012 and Pembrolizumab in Subjects With Advanced HCC	•Biological: GNOS-PV02 •Biological: INO-9012 •Drug: Pembrolizumab •Device: CELLECTRA^®^2000 EP Device	1&2	•Adverse events as graded by CTCAE v5.0; •Immunogenicity of a personalized neoantigen DNA vaccine as measured by interferon-γ secreting T lymphocytes in peripheral blood mononuclear cells (PBMCs) using ELISpot; •Immunogenicity of a personalized neoantigen DNA vaccine as measured by T-cell activation and cytolytic cell phenotype in PBMCs using Flow Cytometry.	NCT04912765 [73]	60	2020
“Cocktail” Therapy for Hepatitis B Related Hepatocellular Carcinoma	•Drug: Cyclophosphamide •Biological: Multiple Signals loaded Dendritic Cells Vaccine	2	•Progression-Free-Survival (PFS) month; •serum AFP (alpha fetoprotein, ng/mL); •serum PIVKA-II (Protein Induced by Vitamin K Absence or Antagonist-II), μg/L.	NCT04317248 [74]	600	2020
Personalized Neoantigen Peptide-Based Vaccine in Combination With Pembrolizumab for the Treatment of Advanced Solid Tumors, The PNeoVCA Study	•Drug: Cyclophosphamide •Biological: Neoantigen Peptide Vaccine •Biological: Pembrolizumab •Biological: Sargramostim	1	•Incidence of adverse events; •The number and percentage of participants who completed the sequencing with satisfactory data quality registration and identified at least 10 actionable peptides, meet the eligibility criteria for registration, and were able to initiate vaccine production; •Immunogenicity responders.	NCT05269381 [75]	36	2022
FusionVAC22_01: DNAJB1-PRKACA Fusion Transcript-based Peptide Vaccine Combined With Immune Checkpoint Inhibition for Fibrolamellar Hepatocellular Carcinoma and Other Tumor Entities Carrying the Oncogenic Driver Fusion	Biological: DNAJB1-PRKACA Fusion Transcript-based Peptide Vaccine	Not specified	Not specified.	NCT05937295 [76]	20	2023
An Open Label, Single-arm, Phase II Neoantigen (NA) Dendritic Cell (DC) Vaccine and Anti-PD1 (Nivolumab) as Adjuvant Treatment in Resected Hepatocellular Carcinoma (HCC) (Group A) and Liver Metastases From Colorectal Cancer (CRLM) (Group B)	Biological: Neoantigen Dendritic Cell Vaccine	Phase II	Not specified.	NCT04912765 [72]	60	2021
Phase 1 Trial of Intravenous Administration of TAEK-VAC-HerBy Vaccine Alone and in Combination With HER2 Antibodies in Patients With Advanced Cancer.	Biological: TAEK-VAC-HerBy Vaccine	Phase I	Not specified.	NCT04246671 [77]	55	2020
Phase I Clinical Trial of Alfa-Fetoprotein,Glypican-3 Based Personalized Cancer Vaccine in Egyptian Patients With Hepatocellular Carcinoma: Pilot Study	Biological: Alfa-Fetoprotein,Glypican-3 Based Personalized Cancer Vaccine	Phase I	Not specified.	NCT05059821 [70]	10	2021
Study of Intratumoral Injection of Dendritic Cells After High-Dose Conformal External Beam Radiotherapy in Patients With Unresectable Liver Cancer	Biological: Dendritic Cells	Not specified	Not specified.	NCT03942328 [78]	54	2023
An Exploratory Study on the Safety and Effectiveness of Autoimmune Cell Therapy Sensitized With Liver Cancer Neoantigen in Patients With High Risk of Recurrence After Surgical Resection of Primary Hepatocellular Carcinoma	Biological: Autoimmune Cell Therapy	Not specified	Not specified.	NCT05105815 [79]	23	2021
A Phase I Pilot Study of Personalized Neoantigen Peptide-Based Vaccine in Combination With Pembrolizumab in Advanced Solid Tumors (PNeoVCA)	Biological: Neoantigen Peptide-Based Vaccine	Phase I	Not specified.	NCT05269381 [75]	36	2022
A Phase I Study of mRNA Vaccine for Patients With HBV-positive Advanced Hepatocellular Carcinoma	Biological: mRNA Vaccine	Phase I	Not specified.	NCT05738447 [57]	9	2023

* This information is available on https://clinicaltrials.gov/, (accessed on 2 August 2023).

## 4. Immunization Approach for Regulating the Tumor Immune Microenvironment (TIME) in Hepatic Cellular Carcinoma

The immune landscape within HCC significantly impacts tumor growth, progression, and treatment response [80]. This landscape comprises various cell types, including cancer, immune, stromal, and endothelial cells, all interconnected in a complex network of cytokines, chemokines, and growth factors [81]. Interactions among these immune cells greatly influence HCC behavior, making them an attractive target for therapeutic interventions [82]. 

Cancer vaccines represent an immunotherapy approach that shifts this landscape towards a more antitumor state [32]. These vaccines primarily aim to activate the immune system, particularly T cells, to recognize and eliminate tumor cells, thus altering the immune responses within the tumor microenvironment [83]. Cancer vaccines can enhance antitumor immunity by increasing the activity of cytotoxic T lymphocytes (CTLs) and natural killer (NK) cells, the immune system’s first line of defense against cancer [84].

However, the HCC microenvironment often exhibits immunosuppressive features that can compromise the effectiveness of immune responses [85]. For example, regulatory T cells (Tregs) and myeloid-derived suppressor cells (MDSCs) within the tumor milieu can inhibit the activity of CTLs and NK cells [86]. Cancer vaccines can potentially counteract these suppressive effects by stimulating the production of immune-enhancing cytokines and inhibiting the activity or presence of immunosuppressive cells [87]. Vaccination may also offer the advantage of fostering long-lasting immune memory [88]. Memory T cells generated post-vaccination can provide extended surveillance and a rapid response upon future encounters with HCC cells, potentially preventing tumor recurrence and contributing to long-term disease management [37].

## 5. Exploring the Synergistic Potential of Cancer Vaccines and Other Immunotherapies in Hepatocellular Carcinoma

While effective on their own, cancer vaccines can also be integrated with other immunotherapeutic strategies to enhance further their impact on the HCC environment [89]. For instance, immune checkpoint inhibitors (ICIs) can be used with vaccines to create a more potent anti-tumor response. While vaccines work to bolster the immune system’s attack on the tumor, ICIs function by dismantling the defense mechanisms that cancer cells use to evade immune cells, thus creating a more favorable environment for anti-tumor immunity [90].

In essence, the potential for cancer vaccines to recalibrate the immune landscape within HCC offers a promising avenue for improving the management of this formidable disease [91]. By amplifying anti-tumor immune responses and counteracting immunosuppressive factors, cancer vaccines could potentially transform the HCC microenvironment, leading to improved patient outcomes [92]

## 6. Future Perspectives

The future of cancer immunotherapy in HCC treatment is marked by considerable promise, with numerous research avenues to explore and potential breakthroughs on the horizon. Developing innovative and more effective cancer vaccines remains at the forefront of this pursuit. Personalized and combination therapies will play a crucial role in enhancing the efficacy of cancer vaccines [93]. Researchers should tailor vaccines to each patient’s unique tumor characteristics, addressing tumor heterogeneity challenges. Furthermore, integrating different cancer vaccine types with other immunotherapeutic modalities, such as immune checkpoint inhibitors and adoptive cell therapies, could amplify antitumor immune responses, leading to improved clinical outcomes [94].

Identifying and targeting specific and immunogenic TAAs or neoantigens is critical for the success of cancer vaccines. Developing novel antigen delivery platforms, like lipid nanoparticles or viral vectors, may enhance vaccine potency and precision, resulting in more potent immune responses and better treatment outcomes.

Overcoming tumor-induced immunosuppression and enhancing the immune system’s ability to recognize and eliminate cancer cells are essential for the efficacy of cancer vaccines. Future research should investigate innovative immune modulatory agents and adjuvants that can synergize with cancer vaccines, promoting more effective and lasting antitumor responses [95].

Predictive biomarkers and models are crucial for optimizing cancer vaccine therapy [96]. Researchers can better determine which patients are most likely to benefit from vaccine therapies by identifying predictive biomarkers and establishing patient selection and response monitoring models. This approach will also enable early treatment response or resistance detection, allowing for more precise therapeutic interventions.

Finally, accelerating cancer vaccine development and approval for HCC treatment requires extensive collaboration among academia, industry, and regulatory agencies. This cooperation should encompass the design of more robust preclinical models, optimization of clinical trial designs, and incorporation of real-world evidence to better understand the safety, efficacy, and long-term benefits of cancer vaccines in diverse patient populations.

Finally, the continued development of innovative cancer vaccines can potentially transform the HCC treatment landscape. As our understanding of tumor biology and the immune system’s role in cancer progression deepens, more effective and personalized cancer vaccines become increasingly feasible. By addressing the challenges and capitalizing on the opportunities outlined here, researchers and clinicians can collaborate to develop groundbreaking cancer vaccines, ultimately improving the prognosis and quality of life for individuals living with HCC.

## 7. Conclusions

In conclusion, cancer treatment vaccines present substantial potential for the successful management and possible prevention of HCC, the most common type of liver cancer globally. Immunotherapy has catalyzed recent progress in personalized cancer vaccines, with strategies rooted in peptides, dendritic cells, viral vectors, and mRNA showing promise in creating targeted immune responses against HCC. Despite intrinsic challenges, such as identifying tumor antigens and the clonal heterogeneity within tumors, cancer vaccines demonstrate many strong opportunities for refining HCC treatment approaches. Research initiatives must continue, particularly focusing on clinical trials, to assess these vaccines’ safety, effectiveness, and clinical relevance among HCC patients. Our aspiration is that with enhanced cancer vaccination protocols we can attain long-term tumor control and enduring remission, ultimately improving prognosis outcomes for individuals battling HCC.

## Data Availability

Not applicable.

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
