# Peer review of "A Comprehensive Review on Cancer Vaccines and Vaccine Strategies in Hepatocellular Carcinoma"

_vaccines, 2023, doi:10.3390/vaccines11081357_

Round 1

Reviewer 1 Report

Estimated Editors in Vaccines,

I've read with great interest the paper from TOJJARI et al. reporting on vaccines targeting HCC. Because of the morbidity and mortality associated with HCC, the implementation of preventive interventions targeting this disorder could represents a significant breakthrough from both clinical and public health point of view.

TOJJARI et al, provide an informative summary of available knowledge on this topic but - unfortunately, they largely fail their aims because of some significant shortcomings in the design of this narrative review. As these issues could be fixed by reworking of the main text, I'm confident that Authors could consistently improve the present study, as follows:

1. As Authors correctly report in the section 3, but then fail to address in subsequent sections, neoplasia targeting vaccines could be designed as preventive interventions or as therapeutic ones. Across the main text this dichotomy is not properly addressed as no preventive vaccines are really reported. Please revise your text in order to fill this gap.

2. Authors have focused section 5 of this paper on the ongoing clinical trials. In order to improve the readability, Authors should/could move the reports on ongoing trials when dealing with the assessed vaccine(s). 

3. I've noticed that Table 1 did include some studies on mAb such as Atezolizumab and Bevacizumab. Please note that mAb are not usually acknowledged as "vaccines" and their inclusion should be either properly explained or removed. 

4. Please include the number of patients included in the clinical trials you've included in Table 1.

5. Figure 1 is not really informative, particularly the lower section. Please evaluate its potential removal.

6. Please include a summary table on the subcategories of vaccines, focusing on their pros and cons, in order to improve the comparison between available and forthcoming formulates.

Reviewer 2 Report

This review article aims to delve into the significance of cancer vaccines as a promising therapeutic strategy for hepatocellular carcinoma (HCC), a prevalent and deadly form of liver cancer. By exploring existing literature, authors highlight various types of cancer vaccines, their challenges, and their role in modulating the immune response within HCC. Additionally, they emphasize the importance of personalized therapies, novel antigen delivery platforms, and predictive biomarkers, along with clinical trials and future perspectives in this field.

Strengths:

1.      The authors have demonstrated a thorough and exhaustive citation of recent findings, contributing to a well-documented review, particularly concerning the various types of vaccines and current available clinical trials.

2.      The inclusion of comprehensive tables in the article adds value to the discussion, presenting information in a clear and organized manner.

Limitations:

1.    Paragraph 3: To enhance readability and provide a concise overview, a table summarizing the types of vaccines, their mechanisms, advantages, and limitations should be included.

2.    Paragraph 3: While the authors have covered peptide-based, dendritic cell-based, viral vector-based, DNA, and mRNA vaccines, we suggest adding sections on other vaccine types, such as cell lysates and adoptive cell therapy/vaccines, to ensure a comprehensive analysis.

3.    Paragraph 4: This section appears underdeveloped. To improve clarity, we propose splitting it into two paragraphs. The first should focus on the contribution of vaccines to reshaping the tumor immune microenvironment (TIME), while the second should explore the use of cancer vaccines in combination treatments.

4.    I understand that perspectives is mostly an opinion emerging from available research. However, they should still be supported by references. Therefore, related references are needed for these sections.

Round 2

Reviewer 1 Report

Estimated Authors,

I've appreciated your efforts to cope with my recommendations.

The paper has been properly improved and I've no further requests for you.

The English text Is now quire appropriate.